# Transcription and Metabolic Profiling Analysis of Three Discolorations in a Day of *Hibiscus mutabilis*

**DOI:** 10.3390/biology12081115

**Published:** 2023-08-10

**Authors:** Zhangshun Zhu, Xinmei Zeng, Xiaoqing Shi, Jiao Ma, Xiaoli Liu, Qiang Li

**Affiliations:** 1Chengdu Botanical Garden (Chengdu Park Urban Plant Science Research Institute), Chengdu 610083, China; zzs19661104@163.com (Z.Z.); zxm2010709@163.com (X.Z.); shixiaoqing87@126.com (X.S.); mj20230621@163.com (J.M.); 2School of Food and Biological Engineering, Chengdu University, Chengdu 610106, China

**Keywords:** flower color transition, mechanism, breeding, gardening, metabolic pathway, gene regulation

## Abstract

**Simple Summary:**

This study revealed the internal regulation mechanism of *Hibiscus mutabilis* discoloration three times a day through joint Transcriptome and metabonomics, which promoted the molecular breeding of ornamental plants.

**Abstract:**

In this study, we used combined transcriptomics and metabolomics to analyze the *H. mutabilis* cultivar’s genetic and physiological mechanisms during three flower color transition periods (from white to pink, then from pink to red) within the span of one day. As a result, 186 genes were found to be significantly increased with the deepening of the *H. mutabilis* flower color; these genes were mainly involved in the expression of peroxidase 30, zinc finger protein, phosphate transporter PHO1, etc. In contrast, 298 genes were significantly downregulated with the deepening of *H. mutabilis* flower color, including those involved in the expression of probable O-methyltransferase 3, copper binding protein 9, and heat stress transcription factor A-6b. Some genes showed differential expression strategies as the flower color gradually darkened. We further detected 19 metabolites that gradually increased with the deepening of the *H. mutabilis* flower color, including L-isoleucine, palmitic acid, L-methionine, and (+)-7-isonitrobenzene. The content of the metabolite hexadecanedioate decreased with the deepening of the *H. mutabilis* flower color. Combined transcriptomics and metabolomics revealed that the metabolic pathways, including those related to anthocyanin biosynthesis, cysteine and methionine metabolism, and sulfur metabolism, appear to be closely related to *H. mutabilis* flower color transition. This study served as the first report on the genetic and physiological mechanisms of short-term *H. mutabilis* flower color transition and will promote the molecular breeding of ornamental cultivars of *H. mutabilis*.

## 1. Introduction

*Hibiscus mutabilis* is an important ornamental plant from the Malvaceae family that originated in China and has been cultivated in China for more than 2000 years [1]. As an ornamental plant, *H. mutabilis* has spread to Japan, Korea, and other countries in Asia. Species of the Malvaceae family are considered to have great economic and ornamental value, and include species such as *Gossypium hirsutum* and *Hibiscus syriacus* (Rose of Sharon). The former is the main global source of natural textile fiber production [2], and the latter is the national flower of Korea, and has attractive white, pink, red, lavender, or purple flowers and a long bloom period [3]. *H. mutabilis* is also studied worldwide because of its attractive flowers, which have a long bloom period, generally 3–4 months, although the persistence of individual flowers lasts for fewer than 48 h. Unlike some other ornamental plants, such as chrysanthemums [4], *Narcissus pseudonarcissus* [5], and *Brassica napus* [6], different flower colors exist among different cultivars. Some *H. mutabilis* cultivars can show rapid flower color change on the same day. The most famous cultivar is ‘Drunk Girl’, also known as ‘Zui Furong’, which can display three different colors over the course of one day, from white in the morning to pink at noon and dark red in the evening. This obvious color change has attracted a large number of visitors from around the world. *H. mutabilis* bears important cultural and historical significance in the city of Chengdu, and is regarded as the city flower.

In addition to being an ornamental plant, *H. mutabilis* is also used as a source of traditional medicine in China; its functions of detoxification, detumescence, and pain relief have led to it being used for a long time to treat ulcers, swelling, herpes zoster, burns, and bruises [7,8,9,10]. Modern medical research has shown that the main active ingredients in *H. mutabilis* include flavonoid glycosides, ferulic acid, lectin, rutin, and isoquercetin, which have antioxidant, anti-insect, anti-viral, and anti-inflammatory effects [11,12,13,14]. The flowers of *H. mutabilis* have also been shown to contain a variety of bioactive substances, such as flavonoids, rutin, and quercetin [13]. Despite being an important ornamental and medicinal plant, the mechanism of flower color change in *H. mutabilis* is still unknown. The gene regulation mechanisms of flower color change and of the change in metabolites need to be further analyzed.

In this study, we collected *H. mutabilis* flower samples during three color transition periods and then used transcriptomics and metabolomics to reveal the gene regulation and metabolic mechanism of the *H. mutabilis* color transition process. The research results revealed the genetic and physiological mechanisms of *H. mutabilis* color transition during three time periods in a day, which will provide guidance for the molecular breeding of ornamental plants and the development or utilization of natural active ingredients in *H. mutabilis*.

## 2. Materials and Methods

### 2.1. Sample Collection

We collected different colors of *H. mutabilis* flowers from one *H. mutabilis* strain in the Chengdu Botanical Garden in Sichuan, China. During one day of the *H. mutabilis* bloom period, we collected 6 white flowers at 8:00, 6 pink flowers at 13:00, and 6 red flowers at 17:00, all from the same plant (Figure 1). The collected flowers were stored on dry ice and then returned to the laboratory for subsequent RNA and metabolite extraction. White *H. mutabilis* flowers were assigned the code ZFRW, pink ones the code ZFRP, and red ones the code ZFRR in subsequent analyses.

### 2.2. RNA Preparation and Library Construction

The total RNA of collected flowers was extracted using a plant RNA Kit (Omega Bio-Tek, Norcross, GA, USA). The extracted RNA was monitored on 1% agarose gels. The RNA concentration was measured using a Qubit^®^ RNA Assay Kit and a Qubit^®^ 2.0 Fluorometer (Life Technologies, Carlsbad, CA, USA). A total amount of 1.5 µg of RNA per sample was used as the input material for RNA sample preparation. We constructed RNA sequencing libraries using the NEBNext^®^ Ultra™ RNA Library Prep Kit for Illumina^®^ (NEB, Ipswich, MA, USA) according to the manufacturer’s instructions. The quality of the obtained RNA sequencing libraries was assessed with the Agilent Bioanalyzer 2100 system (Agilent Technologies, Santa Clara, CA, USA). The libraries were then sequenced on an Illumina HiSeq platform, and paired-end reads were generated.

### 2.3. Quality Control and Transcriptome Assembly

The obtained paired-end reads were first processed through quality control steps to generate clean reads. Briefly, raw reads with low quality and containing adapter sequences or poly-N stretches were removed using Perl scripts. We further calculated the Q20, Q30, and GC contents and sequence duplication level of the clean data. All downstream analyses were based on clean data with high quality. We used Trinity to assemble the transcriptome with min_kmer_cov set to 2 by default and all other parameters set to default [15]. The assembled transcripts were functionally annotated according to seven databases, including Nr (NCBI nonredundant protein sequences) [16], Nt (NCBI nonredundant nucleotide sequences) [16], Pfam (Protein family) [17], KOG/COG (Clusters of Orthologous Groups of proteins) [18], Swiss-Prot (a manually annotated and reviewed protein sequence database) [19], KO (KEGG Ortholog database) [20], and GO (Gene Ontology) [21].

### 2.4. Differentially Expressed Gene Analysis

Clean data were first mapped back onto the assembled transcriptome. Read counts for each gene were obtained from the mapping results. We calculated the gene expression level for each sample using RSEM [22]. We performed differential expression analysis using the DESeq R package [23]. The resulting *p* values were adjusted using Benjamini and Hochberg’s approach for controlling the false discovery rate. Genes with an adjusted *p*-value of <0.05 were considered differentially expressed.

Gene ontology (GO) enrichment analysis of the differentially expressed genes (DEGs) was conducted with the GOseq R package using the default Wallenius noncentral hypergeometric distribution [24]. The statistical enrichment of differentially expressed genes in KEGG (Kyoto Encyclopedia of Genes and Genomes) pathways was tested using KOBAS 3.0 software [25].

### 2.5. Extraction of Metabolites

Approximately 200 mg of a petal sample was added to a 2 mL Eppendorf tube, and 0.6 mL of 2-chlorophenylalanine (4 ppm) methanol (−20 °C) was then added and vortexed for 30 s. The mixture was ground using a tissue grinder (DHTISSUE-32, LAWSON, Ningbo, China) for 90 s at 55 Hz. The ground plant tissue mixture was sonicated at room temperature for 15 min. We then centrifuged the mixture at 12,000 rpm for 10 min and took 200 μL supernatant by using 0.22 μM filter membrane filtration, and the collected filter solution was used for subsequent detection. All of the samples were randomly injected into the apparatus to avoid system errors. We also conducted quality control (QC) to ensure the stability of the system.

### 2.6. Metabolome Detection

We used a Thermo Ultimate 3000 system and an ACQUITY UPLC^®^ HSS T3 (150 × 2.1 mm, 1.8 µm, Waters, Milford, MA, USA) column for the chromatographic detection of the extracted metabolites. The temperature of the automatic sampler was set to 8 °C, the flow rate was 0.25 mL/min, the column temperature was 40 °C, and the sample was injected at 2 μL for gradient elution. The ESI-MSn experiments were executed on a Thermo Q Exactive Focus mass spectrometer with spray voltages of 3.5 kV and −2.5 kV in positive and negative modes, respectively. The detection conditions of metabolites were set according to those in previously published papers.

### 2.7. Metabolite Data Analysis

We used Waters Masslynx 4.1 software to extract metabolite peaks from the LC–MS/MS data. Proteowizard software (v3.0.8789) was then used to turn the raw MS files into the mzXML format, which were sequentially processed using XCMS software for chromatographic matching, metabolic feature detection, and metabolite peak alignment. The metabolite peak areas were normalized by the sum of the areas and subjected to statistical analyses.

Soft Independent Modeling of Class Analogy (SIMCA)-P (version 11.0, Umetrics AB, Umeå, Sweden) was used to conduct multivariate analyses. All variables were UV (Unit Variance) scaled before partial least squares-discriminant analysis (PLS-DA). We used a loading plot to show how different variables contributed to the samples. Based on the loading plot in PLS-DA, notably different metabolites were identified. Afterward, *t*-tests were performed independently to determine significant metabolites (*p* ≤ 0.05) between the groups. An analysis of pathway topology on Metaboanalyst [26] and the KEGG metabolic database [20] was conducted to map metabolic pathways.

The Pearson correlation coefficient was used to analyze the correlation between metabolite expression and transcriptional expression. A metabolic pathway was drawn and visualized using Cytoscape 3.02 [27].

## 3. Results

### 3.1. Overview of the Transcriptome Data

To study the gene regulation mechanism of *H. mutabilis* flower color change over the course of one day, we analyzed the gene expression profiles of *H. mutabilis* flowers from three color transition periods by transcriptome sequencing. On average, 6.52 Gbp of raw RNA data was obtained per sample, with a Q20 and Q30 of 97.78% and 93.84%, respectively (Appendix A). We conducted quality control steps to remove reads with low quality from the raw RNA reads, and approximately 11.20% of the raw reads were discarded. An average of 38,570,298 clean reads per sample were used for subsequent analysis. The obtained clean reads were used to assemble *H. mutabilis* transcripts and unigenes. A total of 352,980 transcripts and 116,127 unigenes were obtained, with total lengths of 352.11 Mbp and 95.91 Mbp, respectively (Appendix A). The GC contents of the obtained transcripts and unigenes were 41.32% and 40.64%, respectively. We further annotated the *H. mutabilis* unigenes against the seven databases (Appendix A); 62.75% of the unigenes could be annotated in the NR database, 58.44% could be annotated in the eggNOG database, and a total of 64.46% could be annotated in at least one of the seven databases.

### 3.2. Gene Expression Analysis

We analyzed the expression profiles of *H. mutabilis* in different color transition periods and analyzed the correlation between the expression profiles of different samples (Figure 2). The results showed that *H. mutabilis* samples from the same color transition period had a greater correlation, while the expression profiles from different color transition periods showed significant differences. The expression profiles of white and pink *H. mutabilis* were more similar to each other than they were to those of red samples. The analysis of differentially expressed genes showed that 16,451 differentially expressed genes were found between white and pink flowers, including 8757 upregulated genes and 7694 downregulated genes in pink flowers relative to white flowers (Figure 3). There were 8484 differentially expressed genes in pink and red *H. mutabilis* flowers, including 4206 upregulated genes and 4278 downregulated genes in red *H. mutabilis* flowers relative to those in pink *H. mutabilis* flowers. We detected 13,323 specific differentially expressed genes between the white and pink *H. mutabilis* flower samples and 5356 specific differentially expressed genes between the pink and red *H. mutabilis* flower samples (Figure 4). A total of 3128 common differentially expressed genes were detected between the white and pink *H. mutabilis* samples and between the pink and red *H. mutabilis* flower samples.

### 3.3. Genes Upregulated with Darker Flower Color

Statistical analysis showed that the expression of 186 genes significantly increased with the deepening of *H. mutabilis* flower color. These mainly included genes such as those involved in the expression of peroxidase 30, zinc finger protein, phosphate transporter PHO1, cytochrome P450 82c4, basic leucine zipper 1, peroxidase 54, oligopeptide transporter 4, and cytochrome c1 (Appendix A).

In addition, there were 4389 genes that were upregulated in the process of *H. mutabilis* flowers transitioning from white to pink, but there was no significant change in the expression level during the transition from pink to red. The genes upregulated in the transition from white to pink included those involved in the expression of zinc finger homeodomain protein, fumarylacetase, malate dehydrogenase, methylenetetrahydrofolate reductase 2, transcription reporter myb5, ATP-dependent zinc metalloproteinase FtsH, probable acyl activating enzyme 2, probable transcription factor kan2, high-affinity nitrate transporter 2.5, calcium-dependent protein kinase 9, and flavanone 3-dioxygenase 1.

### 3.4. Genes Downregulated with Darker Flower Color

There were 298 genes that were significantly downregulated with the deepening of *H. mutabilis* flower color, such as genes that are mainly involved in the expression of probable O-methyltransferase 3, copper binding protein 9, fatty acid biosynthesis 1 isoform 1, heat stress transcription factor A-6b, heat stress transcription factor C-1, auxin response factor 7, molybdenum cofactor sulfurase, glutamate formimidoyltransferase, and short-chain dehydrogenase reductase 2a (Appendix A).

We detected 3388 genes that were first upregulated during the *H. mutabilis* flower color transition from white to pink and that then showed no significant change during the transition from pink to red. The upregulated genes included those involved in the expression of auxin response factor 11, metal tolerance protein 4, anthranilate synthase beta subunit 2, probable methyltransferase PMT28, probable inactive receptor kinase At4g23740, mini zinc finger protein 2, signal peptide peptidase-like 2, and ubiquinone biosynthesis COQ4.

### 3.5. Genes Differentially Expressed with Darker Flower Color

The expression levels of 114 genes increased first and then decreased as the flowers of *H. mutabilis* changed from white to pink and then to red. These differentially expressed genes included those involved in the expression of chalcone synthase 2, protochlorophyllide reductase, oxygen-evolving enhancer protein 1, photosystem I reaction center subunit XI, UDP-glycosyltransferase 74B1, homeobox-leucine zipper protein ATHB-6, and probable metal-nicotianamine transporter YSL5 (Appendix A).

In addition, a total of 9 genes significantly decreased and then increased as the flowers of *H. mutabilis* changed from white to pink and then to red. These genes included those involved in the expression of meiotic nuclear division protein 1, arginine decarboxylase, WD repeat-containing 62, organic cation/carnitine transporter 3, plant cysteine oxidase 1, aquaporin PIP2-1, and haloacid dehalogenase-like hydrolase superfamily protein.

We also detected 658 genes that showed no significant change during the *H. mutabilis* flower color transition from white to pink but that were then significantly downregulated during the transition from pink to red. These genes included those involved in the expression of photosystem I reaction center subunit V, ribulose bisphosphate carboxylase small chain, serine hydroxymethyltransferase, probable sulfate transporter 3.5, protochlorophyllide reductase, photosystem I subunit O, homeobox-leucine zipper protein HDG1, and serine protease SPPA.

### 3.6. GO Functional Enrichment Analysis

We further conducted a GO enrichment analysis of the DEGs between the white and pink *H. mutabilis* flower samples (Figure 5a). The results showed that molecular functions related to oxidoreductase activity, catalytic activity, and transporter activity; cellular components related to the extracellular region; and biological processes related to the oxidation—reduction process, aromatic amino acid family metabolic process, and inorganic anion transport were all significantly enriched in the pink *H. mutabilis* flower samples compared with those in the white flower samples. In addition, the results showed that cellular components related to the membrane and its intrinsic and integral components, and that biological processes related to responses to auxin, cell wall organization, and external encapsulating structure organization, were significantly decreased in the pink *H. mutabilis* flower samples compared with those in the white flower samples.

In the process of *H. mutabilis* changing from pink to red, some cellular components and molecular functions were significantly enhanced. This included cellular components, such as those related to the ribosomal subunit, the ribosome, and the cytosolic ribosome, and molecular functions, such as those related to the structural constituents of the ribosome and structural molecule activity (Figure 5b). However, molecular functions related to oxidoreductase activity, tetrapyrrole binding, and cofactor binding were significantly reduced. Biological processes related to the auxin-activated signaling pathway, the cellular response to auxin stimulus, and the response to auxin were also significantly reduced, as were cellular component DEGs related to the thylakoid, photosynthetic membrane, and chloroplasts.

### 3.7. KEGG Pathway Enrichment Analysis

KEGG pathway enrichment analysis for the *H. mutabilis* flower samples showed that, compared with white flower samples, the pink flower samples were significantly enriched in pathways related to phenylpropanoid biosynthesis; flavonoid biosynthesis; phenylalanine metabolism; phenylalanine, tyrosine, and tryptophan biosynthesis; glycerolipid metabolism; starch and sucrose metabolism; and tryptophan metabolism (Figure 6a). Compared with the white flower samples, the pink flower samples were significantly decreased in pathways related to plant hormone signal transduction, steroid biosynthesis, fatty acid elongation, and cutin, suberine, and wax biosynthesis.

When the *H. mutabilis* flowers changed from pink to red, the pathways were significantly enhanced for ribosome, pentose, and glucuronate interconversions, as were the pathways for oxidative phosphorylation. The KEGG pathways that were significantly decreased included those related to plant hormone signal transduction; photosynthesis—antenna proteins; photosynthesis; tyrosine metabolism; isoquinoline alkaloid biosynthesis; flavonoid biosynthesis; tropane, piperidine, and pyridine alkaloid biosynthesis; the pentose phosphate pathway; and phenylalanine metabolism (Figure 6b).

### 3.8. Metabolomic Profiles of H. mutabilis Flowers

We used the metabolome method to analyze the metabolic profiles of the flowers of *H. mutabilis* from the three color transition periods, and a total of 273 metabolites were detected. The OPLS-DA clearly distinguished the different flower colors of *H. mutabilis* samples according to their metabolic profiles, showing that there were significant differences in the metabolites of the three different *H. mutabilis* color samples (Figure 7).

### 3.9. Differentially Expressed Metabolites

The statistical analysis revealed that the expression of 19 metabolites gradually increased with the deepening of the *H. mutabilis* flower color. These metabolites included L-isoleucine, palmitic acid, L-methionine, (+)-7-isocyanmonic acid, acetylcholine, 3,4-dihydroxybenzeneacetic acid, 2-keto-6-acetamidocaproate, and 3-O-methylgallate. The content of one metabolite gradually decreased with the deepening of the *H. mutabilis* flower color, and that metabolite was hexadecanedioate (Figure 8).

A total of 20 metabolites in the *H. mutabilis* flowers showed no significant change in expression during the transition from white to pink. However, the expression of these 20 metabolites increased significantly during the transition from pink to red. These metabolites included nalbuphine, trans-trans-muconic acid, 2-isopropylmalic acid, guanidinosuccinic acid, D-glucuronic acid, itaconic acid, and stearic acid. In addition, 20 metabolites in *H. mutabilis* showed no significant change in expression during the transition from white to pink, while the expression decreased significantly during the transition from pink to red. These 20 metabolites included guanidoacetic acid, pyruvic acid, exemestane, 2-iminobutanoate, phosphoglycolic acid, palmitoleic acid, allantoic acid, 2-keto-glutaramic acid, and 3-dehydroshikimate.

A total of 122 *H. mutabilis* flower metabolites only increased during the transition from white to pink (Figure 9), with no significant change in the expression of these metabolites during the transition from pink to red. These metabolites included 6-hydroxynicotinate, lithocholic acid, nicotinic acid, uric acid, pantothenic acid, 4-hydroxycinnamic acid, gentisic acid, and sinapic acid. The expression levels of four *H. mutabilis* flower metabolites were significantly decreased during the change from white to pink, while the expression levels of these metabolites did not change significantly during the change from pink to red. These four metabolites were guanosine, ketoleucine, 4,5-dihydroorotic acid, and picolinic acid. In addition, the content of six metabolites increased first and then decreased as the *H. mutabilis* flower color became darker. These metabolites were hydroxykynurenine, methyl beta-D-galactoside, cyclic AMP, gamma-aminobutyric acid, 13-L-hydroperoxylinoleic acid, and pyrrole-2-carboxylic acid.

### 3.10. Transcriptome and Metabolome Association Analysis

An association analysis based on the KEGG metabolic pathway analysis showed that anthocyanin biosynthesis, cysteine and methionine metabolism, C5-branched dibasic acid metabolism, glutathione metabolism, and beta-alanine metabolism played key regulatory roles in the *H. mutabilis* flower transition from white to pink (Appendix A). As the *H. mutabilis* flowers changed from pink to red, sulfur metabolism, taurine and hypotaurine metabolism, purine metabolism, cyanoamino acid metabolism, glycine, serine, and threonine metabolism, vitamin B6 metabolism, amino sugar and nucleotide sugar metabolism, and fatty acid metabolism played key regulatory roles.

## 4. Discussion

As the city flower of Chengdu, *H. mutabilis* has important historical, cultural, and ornamental value [1]. *H. mutabilis* is popular with tourists because of its variety and ability to change colors over the course of one day. The most famous variety is *H. mutabilis*, the flowers of which can transition from white to pink and then red in one day. In addition, *H. mutabilis* is also regarded as an important source of Chinese medicine and some natural active substances [28,29,30]. The flowers of many plants, such as chrysanthemum and monkeyflower [31], can show different colors within or between species, and their color changes are considered to be closely related to the anthocyanin content or environmental responses [32,33,34,35]. However, few plants can change the color of their flowers on such a short time scale. To date, the physiological and genetic mechanisms of flower color transition over a short time period have not been analyzed [36,37]. With the development of sequencing and material identification technology, scholars can identify the genetic mechanisms of plant flower color transition through multiomics [38,39,40,41]. This study serves as the first study to analyze the genetic and physiological mechanisms of short-term flower color change in *H. mutabilis* by combining transcriptomics and metabolomics, providing an important reference for the molecular breeding of ornamental cultivars of *H. mutabilis.*

The transcriptional analysis in this study revealed that 186 genes significantly increased with the deepening of *H. mutabilis* flower color, including those related to the expression of peroxidase 30, zinc finger protein, phosphate transporter PHO1, cytochrome P450 82c4, basic leucine zipper 1, peroxidase 54, oligopeptide transporter 4, and cytochrome c1 [42,43]. Zinc finger proteins have been found to play an important role in flower development in various plants [44]. Cytochrome P450 enzymes have been found to play an important role in the biosynthesis of flavonoids and anthocyanins, which are major pigments found in flowers [45,46]. In addition, it has been discovered that basic leucine zipper (bZIP) is a key regulator of the abscisic acid (ABA) signaling pathway controlling seed dormancy, germination, plant growth, and flowering [47,48]. Furthermore, in this study, 298 genes were significantly downregulated with the deepening of *H. mutabilis* flower color, which mainly included genes such as those related to the expression of probable O-methyltransferase 3, copper binding protein 9, fatty acid biosynthesis 1 isoform 1, heat stress transcription factor A-6b, heat stress transcription factor C-1, auxin response factor 7, molybdenum cofactor sulfurase, glutamate formimidoyltransferase, and short-chain dehydrogenase reductase 2a. In flower pigmentation, O-methyltransferase (OMT) plays an important role in the modification of anthocyanins [49,50]. Auxin response factors have been reported to play an important regulatory role in flower development [51,52]. In addition, *H. mutabilis* has developed a variety of gene regulation modes for the short-term flower color change process. Different genes show variable expression patterns at different color transition stages. In general, thousands of genes are involved in the short-term flower color changes of *H. mutabilis*, and some of them play key regulatory roles in flower development and pigment regulation.

Metabolite analysis showed that there were significant differences in metabolite profiles during different *H. mutabilis* flower color transition stages, and the expression of 19 metabolites increased with the deepening of the flower color. These metabolites included L-isoleucine, palmitic acid, L-methionine, (+)-7-isocyanmonic acid, acetylcholine, 3,4-dihydroxybenzeneacetic acid, 2-keto-6-acetamidocaproate, and 3-O-methylgallate. One metabolite gradually decreased with the deepening of the *H. mutabilis* flower color, and that was hexadecanedioate. Interestingly, we did not detect the enrichment of common plant pigments, such as flavonoids and anthocyanins, during the flower color changes of *H. mutabilis* [53,54,55], which showed that the change in *H. mutabilis* flower color could be the result of the interaction of various metabolites [39,56,57]. We also detected metabolites specifically enriched at different flower color transition stages; these metabolites included 2-iminobutanoate, 4-hydroxycinnamic acid, gamma-aminobutyric acid, quercitrin, arachidonic acid, 3-indoleacrylate, palmitic acid, aspartame, 2-oxoarginine, and methyl jasmonate. These natural compounds identified in the *H. mutabilis* flowers have important value in plant growth regulation, the food industry, and medicine, and warrant further research and application [11,13,58].

Based on the combined transcriptome and metabolome, we found that the metabolic pathways, including those related to anthocyanin biosynthesis; cysteine and methionine metabolism; C5-branched dibasic acid metabolism; glutathione metabolism; and beta-alanine metabolism were closely related to the *H. mutabilis* color change from white to pink. Polyphenol compounds, such as anthocyanins, produce the different hues of pink, red, purple, and blue in flowers, vegetables, and fruits [59,60,61,62,63,64,65]. The enrichment of the anthocyanin biosynthesis pathway may play a key role in regulating the color transition of *H. mutabilis* from white to pink [34,66]. In the process of the *H. mutabilis* flowers changing from pink to red, sulfur metabolism, taurine and hypotaurine metabolism, purine metabolism, cyanoamino acid metabolism, glycine, serine, and threonine metabolism, vitamin B6 metabolism, amino sugar and nucleotide sugar metabolism, and fatty acid metabolism played key regulatory roles. In general, in the *H. mutabilis* short-term flower color change regulation, multiple regulatory genes, and different expression patterns of key genes were involved, resulting in the diverse expression of flower metabolites at different color transition stages, which jointly promoted the formation and regulation mechanism of *H. mutabilis* flower color. This study served as the first report on the regulation and physiological mechanism of short-term flower color change in *H. mutabilis*, and will promote the molecular breeding of ornamental cultivars of *H. mutabilis.*

## 5. Conclusions

In this study, a combination of transcriptomics and metabolomics was employed to investigate the genetic and physiological mechanisms of the *H. mutabilis* cultivar’s flower color transition over the course of one day, from white to pink, then from pink to red. The results revealed 186 genes that were significantly upregulated and 298 genes that were significantly downregulated with the deepening of the flower color. These genes were mainly involved in the expression of peroxidase 30, zinc finger protein, phosphate transporter PHO1, probable O-methyltransferase 3, copper binding protein 9, and heat stress transcription factor A-6b. Additionally, 19 metabolites were found to increase with deepening flower color, such as L-isoleucine, palmitic acid, L-methionine, and (+)-7-isonitrobenzene, while the content of hexadecanedioate decreased. The metabolic pathways related to anthocyanin biosynthesis, cysteine and methionine metabolism, and sulfur metabolism were found to be closely associated with the flower color transition. This research serves as the first report on the genetic and physiological mechanisms of short-term *H. mutabilis* flower color transition, and could be beneficial in the molecular breeding of ornamental cultivars of *H. mutabilis*.

## Figures and Tables

**Figure 1 biology-12-01115-f001:**
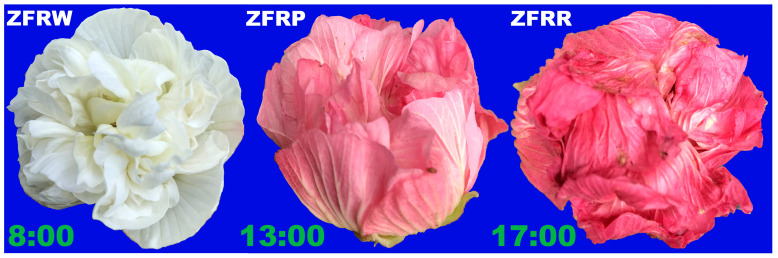
*H. mutabilis* flowers of different colors collected from the same plant at different times of the day. ZFRW, white *H. mutabilis* flowers; ZFRP, pink *H. mutabilis* flowers; ZFRR, red *H. mutabilis* flowers.

**Figure 2 biology-12-01115-f002:**
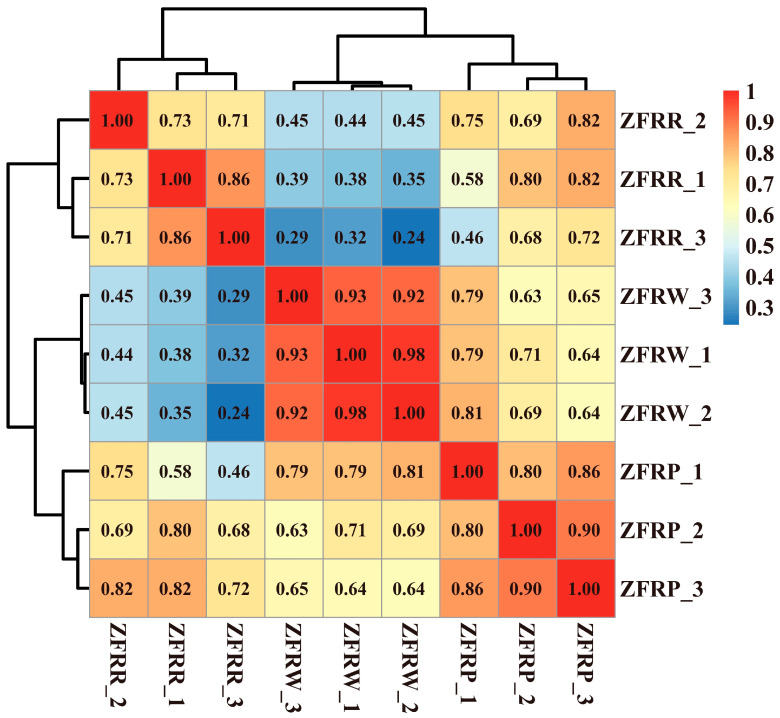
Correlation and cluster analysis of gene expression profiles for *H. mutabilis* flower samples of different colors.

**Figure 3 biology-12-01115-f003:**
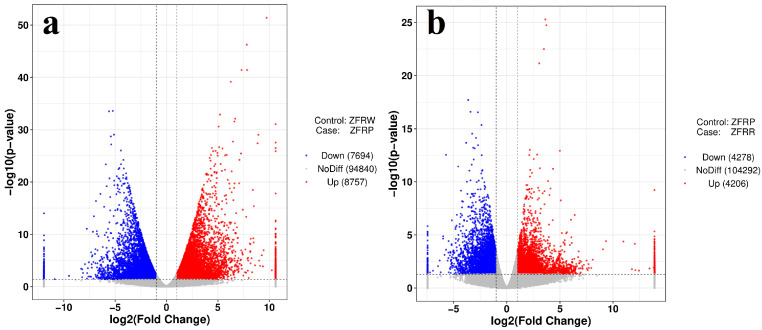
Volcano map of differentially expressed genes between white and pink flowers (**a**) and between pink and red flowers (**b**). ZFRW, white *H. mutabilis* flowers; ZFRP, pink *H. mutabilis* flowers; ZFRR, red *H. mutabilis* flowers.

**Figure 4 biology-12-01115-f004:**
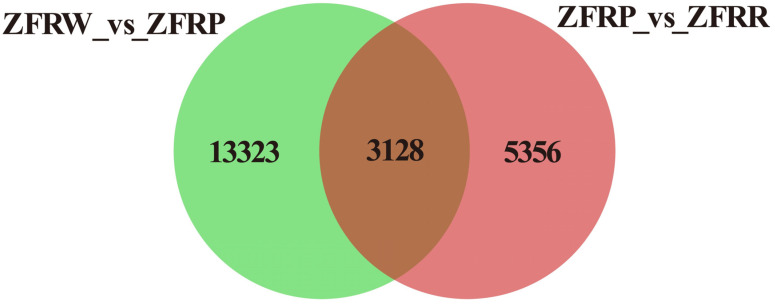
Shared and specific differentially expressed genes between different samples. ZFRW, white *H. mutabilis* flowers; ZFRP, pink *H. mutabilis* flowers; ZFRR, red *H. mutabilis* flowers.

**Figure 5 biology-12-01115-f005:**
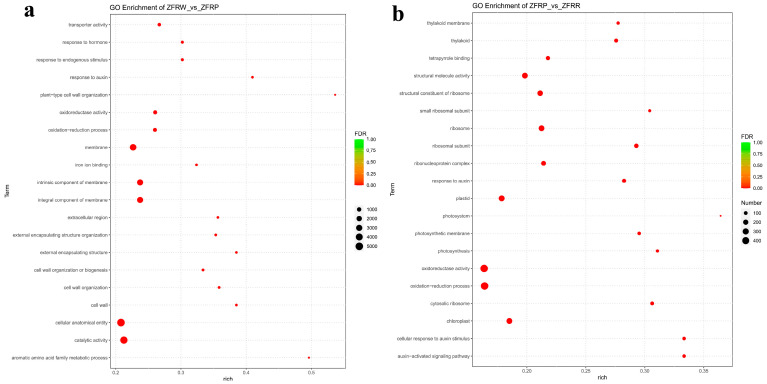
Gene ontology (GO) enrichment analysis of the differentially expressed genes (DEGs) between white and pink flowers (**a**) and between pink and red flowers (**b**).

**Figure 6 biology-12-01115-f006:**
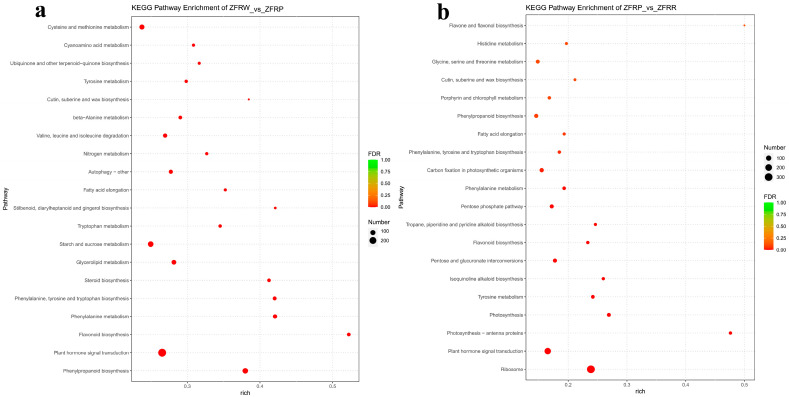
KEGG (Kyoto Encyclopedia of Genes and Genomes) pathway enrichment of differentially expressed genes between white and pink flowers (**a**) and between pink and red flowers (**b**). ZFRW, white *H. mutabilis* flowers; ZFRP, pink *H. mutabilis* flowers; ZFRR, red *H. mutabilis* flowers.

**Figure 7 biology-12-01115-f007:**
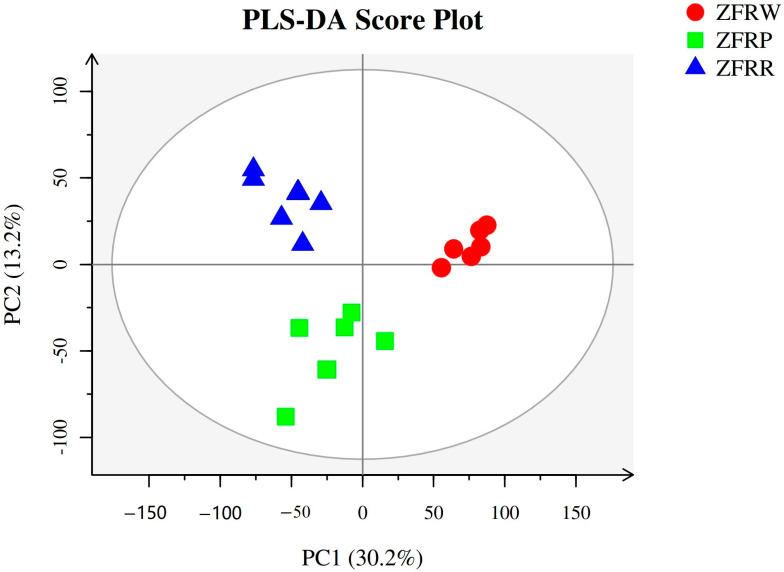
Partial least squares-discriminant analysis (PLS-DA) of metabolic profiles of different samples. ZFRW, white *H. mutabilis* flowers; ZFRP, pink *H. mutabilis* flowers; ZFRR, red *H. mutabilis* flowers.

**Figure 8 biology-12-01115-f008:**
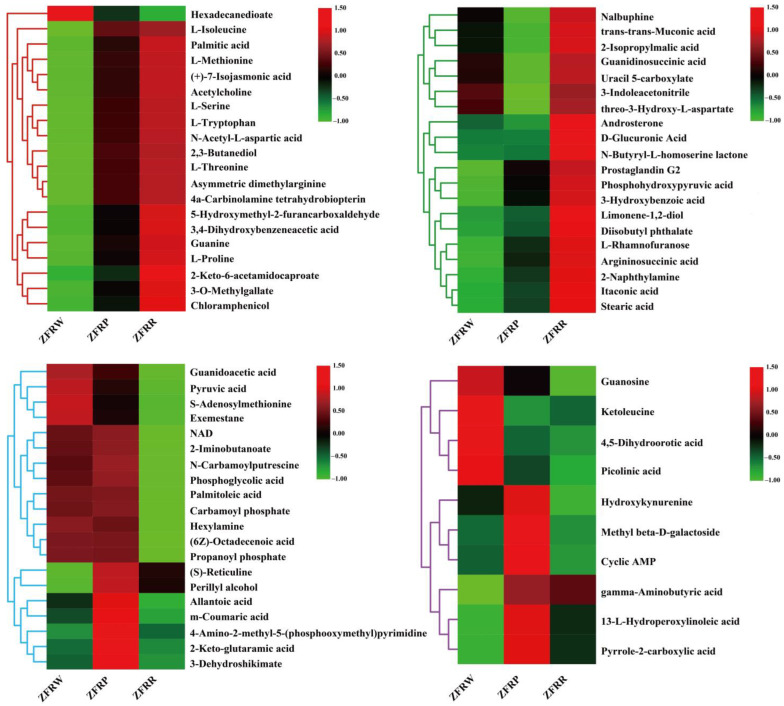
Cluster heatmap analysis of differentially expressed metabolites among different samples. ZFRW, white *H. mutabilis* flowers; ZFRP, pink *H. mutabilis* flowers; ZFRR, red *H. mutabilis* flowers.

**Figure 9 biology-12-01115-f009:**
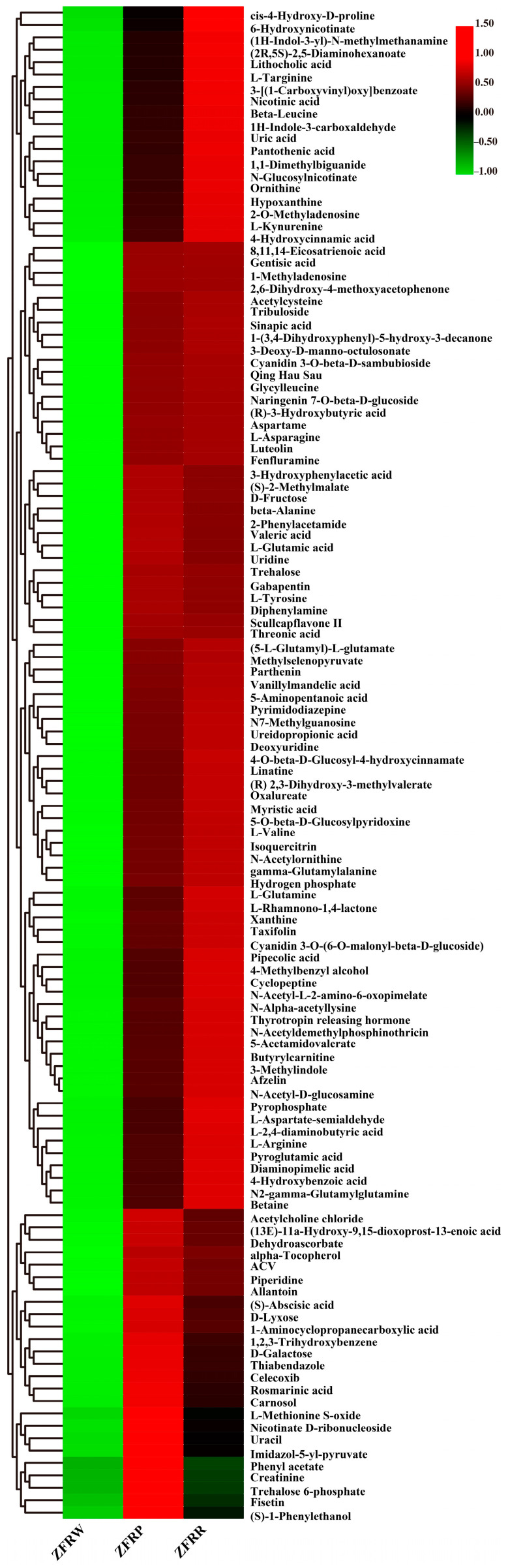
Metabolites increased as *H. mutabilis* flowers transitioned from white to pink, but without a significant change in expression as the flowers changed from pink to red. ZFRW, white *H. mutabilis* flowers; ZFRP, pink *H. mutabilis* flowers; ZFRR, red *H. mutabilis* flowers.

## Data Availability

All data generated or analyzed during this study are included in this published article (and its Appendix A).

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
