# Peer review of "Transcription and Metabolic Profiling Analysis of Three Discolorations in a Day of Hibiscus mutabilis"

_biology, 2023, doi:10.3390/biology12081115_

Round 1
Reviewer 1 Report
Summary:
As few plants can change their flower color over a short period of time, Hibiscus mutabilis (H. mutabilis) is popular with tourists because its flower color can change from white to pink and then to red within the span of one day. However, the mechanisms of such short-term flower color transition have not been elucidated yet. Zhangshun Zhu, et al collected H. mutabilis petal samples at three color stages respectively and then used transcriptomics and metabolomics to reveal the gene expression and metabolic pathways during the H. mutabilis color transition process. The comprehensive data from this study provide knowledge foundation for short-term flower color transition in H. mutabilis, and will promote the breeding of ornamental cultivars of H. mutabilis.
General comments:
The introduction provides sufficient background. Materials & Methods are adequately described. Experimental design is appropriate to pursue the aims of this research and the methodology is sound. The results are clearly presented and precisely interpretated. However, the authors should discuss the relevance and significance of the research to application in terms of how to improve the breeding of H. mutabilis. In addition, the authors also should anticipate the impact this research may have on the breeding of other similar ornamental cultivars.
Author Response
Dear Reviewers:
Thank you for your comments concerning our manuscript. Those comments are all valuable and very helpful for revising and improving our paper. We have studied comments carefully and have made corrections which we hope meet with approval. The main corrections in the paper and the responds to the reviewer’s comments are as flowing:
Summary:
As few plants can change their flower color over a short period of time, Hibiscus mutabilis (H. mutabilis) is popular with tourists because its flower color can change from white to pink and then to red within the span of one day. However, the mechanisms of such short-term flower color transition have not been elucidated yet. Zhangshun Zhu, et al collected H. mutabilis petal samples at three color stages respectively and then used transcriptomics and metabolomics to reveal the gene expression and metabolic pathways during the H. mutabilis color transition process. The comprehensive data from this study provide knowledge foundation for short-term flower color transition in H. mutabilis, and will promote the breeding of ornamental cultivars of H. mutabilis.
Response: Thank you very much for your comments.
General comments:
The introduction provides sufficient background. Materials & Methods are adequately described. Experimental design is appropriate to pursue the aims of this research and the methodology is sound. The results are clearly presented and precisely interpretated. However, the authors should discuss the relevance and significance of the research to application in terms of how to improve the breeding of H. mutabilis. In addition, the authors also should anticipate the impact this research may have on the breeding of other similar ornamental cultivars.
A: The key genes identified in this paper that participate in the discoloration of H. mutabilis provide support for the subsequent targeted breeding of ornamental plant varieties.
Reviewer 2 Report
Comments: The article uses metabolomics and transcriptomics analysis to identify key differences between the three developmental stages. Overall manuscript quality is assessed as good.
Following are few specific comments
Why there is no comparison of secondary metabolites in the paper? Anthocyanin or any other pigments were not detected in the metabolome analysis?
The objective of the study has to be described more effectively/
Are there any previous references interconnecting mechanism of color transition to reported primary metabolism changes? If yes, authors need to add the same.
No information on transcript levels of pigment biosynthesis has been highlighed.
Author Response
Dear Reviewers:
Thank you for your comments concerning our manuscript. Those comments are all valuable and very helpful for revising and improving our paper. We have studied comments carefully and have made corrections which we hope meet with approval. The main corrections in the paper and the responds to the reviewer’s comments are as flowing:
Comments: The article uses metabolomics and transcriptomics analysis to identify key differences between the three developmental stages. Overall manuscript quality is assessed as good.
Response: Thank you very much for your comments.
Following are few specific comments
Why there is no comparison of secondary metabolites in the paper? Anthocyanin or any other pigments were not detected in the metabolome analysis?
Response: Thank you for your suggestions. We analyzed the secondary metabolites, and anthocyanidin was identified during flower discoloration, but its expression did not show significant difference in this study. However, significant differences were found in some other secondary metabolites, including palmitic acid, acetylcholine, 3,4-dihydroxybenzeneacetic acid, etc
The objective of the study has to be described more effectively/
Are there any previous references interconnecting mechanism of color transition to reported primary metabolism changes? If yes, authors need to add the same.
Response: This study served as the first report on the genetic and physiological mechanisms of short-term H. mutabilis flower color transition, and will promote the molecular breeding of ornamental cultivars of H. mutabilis.
No information on transcript levels of pigment biosynthesis has been highlighed.
A: In this study, an association analysis based on the KEGG metabolic pathway analysis showed that anthocyanin biosynthesis played regulatory roles in the H. mutabilis flower transition from white to pink.
Thank you again for your suggestion.
Reviewer 3 Report
Unfortunately, after a careful reading, I cannot recommend Zhu et al. manuscript for consideration in Biology. In my opinion, the manuscript is careless, the content of the article in its current form does not correspond to the title of the article.
Floral color change occurs in H. mutabilis when flowers are white in the morning, turning pink during noon, and red in the evening of the same day. The temperature may be an important factor affecting the rate of color change as white flowers kept in the refrigerator remain white until they are taken out to warm, whereupon they slowly turn pink. In general, scientific papers on this topic are very interesting and meaningful. The problem of color change in this species has been very well studied and described by other authors: Wong, S.K.; Lim, Y.Y.; Chan, E.W.C. (2009). Antioxidant properties of Hibiscus: species variation, altitudinal change, coastal influence and floral colour change. Journal of Tropical Forest Science 21 (4): 307–315. I encourage authors to read this article, it explains a lot…
In general, this manuscript shows obvious deficiencies in all parts. The Introduction must be revised. The Materials and Methods chapter leaves many questions unanswered. Did the authors include temperature effect analysis as a factor in this study? Compared to the scope of the results, the discussion is weak. The sentences from lines 348-364 should be deleted, they are more of an introduction than a discussion. Summarized, the potential of the results of this study is not used for publication. A subchapter Conclusions, summarizing the most important achievements of the authors, should also be added to this article.
Minor:
Title: Please delete the running title
Abstract: Please remove the first sentence of the abstract: “Hibiscus mutabilis is popular with tourists because its flower color can change over a short period of time” - it is inappropriate.
The manuscript needs strong English language checking.
Author Response
Dear Reviewers:
Thank you for your comments concerning our manuscript. Those comments are all valuable and very helpful for revising and improving our paper. We have studied comments carefully and have made corrections which we hope meet with approval. The main corrections in the paper and the responds to the reviewer’s comments are as flowing:
Unfortunately, after a careful reading, I cannot recommend Zhu et al. manuscript for consideration in Biology. In my opinion, the manuscript is careless, the content of the article in its current form does not correspond to the title of the article.
Floral color change occurs in H. mutabilis when flowers are white in the morning, turning pink during noon, and red in the evening of the same day. The temperature may be an important factor affecting the rate of color change as white flowers kept in the refrigerator remain white until they are taken out to warm, whereupon they slowly turn pink. In general, scientific papers on this topic are very interesting and meaningful. The problem of color change in this species has been very well studied and described by other authors: Wong, S.K.; Lim, Y.Y.; Chan, E.W.C. (2009). Antioxidant properties of Hibiscus: species variation, altitudinal change, coastal influence and floral colour change. Journal of Tropical Forest Science 21 (4): 307–315. I encourage authors to read this article, it explains a lot…
Response: Thank you for your suggestion. We have carefully read the paper you provided and cited it in the article. In fact, the external environment, such as temperature, may be one of the regulatory factors that cause the discoloration of hibiscus flowers. However, our aim in this paper is to investigate the internal driving factors for the discoloration of hibiscus flowers, such as genes and metabolites. The external environment such as temperature must also be regulated by regulating the expression of genes or metabolites inside the hibiscus to achieve the effect of regulating the discoloration of hibiscus flowers. The goal of our paper is to analyze the internal driving mechanism of the discoloration of hibiscus flowers and provide reference for targeted breeding of ornamental plants.
In general, this manuscript shows obvious deficiencies in all parts. The Introduction must be revised. The Materials and Methods chapter leaves many questions unanswered. Did the authors include temperature effect analysis as a factor in this study? Compared to the scope of the results, the discussion is weak. The sentences from lines 348-364 should be deleted, they are more of an introduction than a discussion. Summarized, the potential of the results of this study is not used for publication. A subchapter Conclusions, summarizing the most important achievements of the authors, should also be added to this article.
Response: Thank you for your suggestion. The temperature is not within the consideration range of our experiment. We are studying the internal driving mechanism of the color change regulation of the ornamental plant Hibiscus under natural conditions, rather than external environments such as uncontrollable light, temperature, time, humidity, and so on. The external environment inevitably reflects the expression of internal genes or metabolites, and our goal is to reveal these mechanisms. In addition, based on your suggestion, we have added a conclusion to the paper. Thank you.
Minor:
Title: Please delete the running title
Response: The running title have been deleted. Thank you.
Abstract: Please remove the first sentence of the abstract: “Hibiscus mutabilis is popular with tourists because its flower color can change over a short period of time” - it is inappropriate.
Response: We have removed the first sentence of abstract.
The English version of this manuscript has been revised by native English experts.
Thank you again for your suggestion。
Round 2
Reviewer 2 Report
I am still not satisfied with the response about the query: why no pigments were studied in the three developmental stages. Very few phenolics have been detected in the sample analysis.
Experimental work needs to be done in order to justify the title of the manuscript.
minor language editing can be done.
Author Response
Dear Reviewers:
Thank you for your comments concerning our manuscript. Those comments are all valuable and very helpful for revising and improving our paper. We have studied comments carefully and have made corrections which we hope meet with approval. The main corrections in the paper and the responds to the reviewer’s comments are as flowing:
I am still not satisfied with the response about the query: why no pigments were studied in the three developmental stages. Very few phenolics have been detected in the sample analysis.
Response: Several pigments and phenolics have been identified in the flowers of Hibiscus fragrans. In this study, the metabolites displayed in the manuscript were only differentially expressed during the three discoloration stages of Hibiscus mutabilis. The complete metabolic spectrum can be found in the following attachment, which includes multiple pigments and phenolics. Thank you!
Experimental work needs to be done in order to justify the title of the manuscript.
Response: Thank you for your suggestion. I agree with your point of view. Due to the fact that the blooming period of Hibiscus mutabilis is around November each year, it is difficult for us to obtain fresh Hibiscus mutabilis samples for experimental verification at this time. In order to avoid causing confusion, we have changed the title of the manuscript to Transcription and metabolic profiling analysis of three discolourations in a day of Hibiscus mutabilis, making the title of our paper more objectively reflect the research content.
The language of the paper has been rechecked. Thank you again for your suggestion.

Reviewer 3 Report
The authors have adequately addressed my comments and suggestions for improvement and/or correction and significantly improved the manuscript compared to the original version. I have no further comments. I recommend publishing this article.
Author Response
Dear Reviewers:
Thank you for your comments concerning our manuscript.
Round 3
Reviewer 2 Report
Nil